# Acute Zonal Occult Outer Retinopathy in a Patient Suffering from Epilepsy: Five-Year Follow-Up

**DOI:** 10.3390/medicina57111276

**Published:** 2021-11-20

**Authors:** Izabella Karska-Basta, Bożena Romanowska-Dixon, Dorota Pojda-Wilczek, Alina Bakunowicz-Łazarczyk, Agnieszka Kubicka-Trząska, Karolina Gerba-Górecka

**Affiliations:** 1Division of Ophthalmology and Ocular Oncology, Department of Ophthalmology, Faculty of Medicine, Jagiellonian University Medical College, 31-501 Kraków, Poland; romanowskadixonbozena1@gmail.com (B.R.-D.); agnieszka.kubicka-trzaska@uj.edu.pl (A.K.-T.); 2Department of Opthalmology and Ocular Oncology, University Hospital, 31-501 Kraków, Poland; karolina.gerba@gmail.com; 3Department of Ophthalmology, Faculty of Medical Sciences in Katowice, Medical University of Silesia in Katowice, 40-752 Katowice, Poland; pojda-wilczek@wp.pl; 4Department of Pediatric Ophthalmology and Strabismus, Medical University of Białystok, Białystok University Children’s Hospital, 15-274 Białystok, Poland; alina.lazarczyk@umb.edu.pl

**Keywords:** AZOOR, epilepsy, electrophysiology

## Abstract

We report an unprecedented case of a young patient with epilepsy coexisting with acute zonal occult outer retinopathy (AZOOR), a rare white dot syndrome of unknown etiology, associated with damage to the large zones of the outer retina. Recently, it has been established that epileptic episodes contribute to an inflammatory response both in the brain and the retina. A 13-year-old male patient with epilepsy was referred by a neurologist for an ophthalmologic consultation due to a sudden deterioration of visual acuity in the left eye. The examination, with a key role of multimodal imaging including color fundus photography, fluorescein angiography, indocyanine green angiography (ICGA), fundus autofluorescence (FAF), swept-source optical coherence tomography (SS-OCT) with visual field assessment, and electroretinography indicated AZOOR as the underlying entity. Findings at the first admission included enlargement of the blind spot in visual field examination along a typical trizonal pattern, which was revealed by FAF, ICGA, and SS-OCT in the left eye. The right eye exhibited no abnormalities. Seminal follow-up revealed no changes in best corrected visual acuity, and multimodal imaging findings remain unaltered. Thus, no medical intervention is required. Our case and recent laboratory findings suggest a causative link between epilepsy and retinal disorders, although this issue requires further research.

## 1. Introduction

Acute zonal occult outer retinopathy (AZOOR) is a rare clinical entity of unclear etiology, which invariably leads to dysfunction of the outer retinal zones. Typically, AZOOR occurs in young women with myopia. Several causative factors have been implicated in the development of this disorder, including infections and autoimmune diseases [1]. The main symptoms include photopsias and visual field defects, which initially tend to occur unilaterally [1]. Fundoscopic examination is either inconclusive or shows slight abnormalities which do not match presenting symptoms. In those cases, other diagnostic modalities, including fundus autofluorescence (FAF), swept-source optical coherence tomography (SS-OCT), fluorescein angiography (FA), and indocyanine green angiography (ICGA), have a higher sensitivity in diagnosing AZOOR. Regardless of the other imaging modalities, electroretinography (ERG) is crucial for the diagnosis of AZOOR, as it invariably exhibits decreased amplitudes of retinal responses following specific stimuli [1,2].

Epilepsy is a group of diseases characterized by recurrent episodes of seizures [3]. A recent study on rats established that these episodes contribute to an inflammatory response both in the brain and the retina [4], providing some evidence for a possible causative link between epilepsy and retinal disorders. The results, however, need to be confirmed in future research (including epidemiological studies).

## 2. Case Presentation

A 13-year-old male patient with a recent diagnosis of epilepsy was referred by his neurologist to the Department of Ophthalmology and Ocular Oncology at Jagiellonian University Medical College in Kraków, Poland, due to visual deterioration and visual field disturbances in the left eye. The patient denied photopsias. Best corrected visual acuity was 20/20 in the left eye and 20/25 in the right eye (due to anisometropia), while the refractive errors were −2.0/−0.25 axis 173 and −4.25/−0.25 axis 150, respectively. The pupils were equal and exhibited no relative afferent pupillary defect. Intraocular pressure, color vision testing, and slit-lamp biomicroscopy of the anterior segment of the eye indicated no abnormalities, whereas fundus examination revealed only an area of a thinner retina with some hyperpigmentation (Figure 1A). Visual field testing of the right eye was unremarkable but demonstrated enlargement of the blind spot for the left eye (Figure 1C).

The patient had previously been treated at a neurological unit where he was diagnosed with epilepsy on the basis of two unprovoked seizures and epileptic activity on electroencephalography. To prevent any further epileptic seizures, valproate (Depakine) had been administered with satisfactory results. The treatment lasted four years and was discontinued without a relapse of seizures.

Magnetic resonance imaging of the brain and orbits revealed no abnormalities that could explain the symptoms. Routine blood screening as well as titers of specific antibodies (e.g., antinuclear antibodies, antineutrophil cytoplasmic antibodies, antiretinal antibodies) were within reference ranges. Possible infective causes were excluded.

In the left eye, swept-source OCT (SS-OCT) showed peripapillary disruption of the ellipsoid zone (which is associated with photoreceptor damage) (Figure 1E), while SS-OCT of the optic nerve revealed borderline thinning of the retinal nerve fiber layer and ganglion cell complex loss in the temporal quadrant. On FAF, pathological changes in the left eye were detected, including an abnormal area encircled by a rim of high autofluorescence (Figure 1G). Fluorescein angiography revealed peripapillary hyperfluorescence, which persisted throughout all imaging phases (Figure 1I), whereas indocyanine green angiography demonstrated delayed hyperfluorescence at the lesion border with foveal sparing (Figure 2A–C). The findings were consistent with the SS-OCT image and corresponded to the visual field defect. The right eye appeared normal on all of the additional imaging modalities.

Electrophysiological examinations were performed using EP-1000 (Tomey, Japan). All examinations were compliant with the International Society for Clinical Electrophysiology of Vision standards [5,6,7]. Pattern ERG (PERG), flash full-field ERG, and multifocal ERG were recorded with a silver thread (DTL) electrode. Retinal function in the right eye was normal. Flash full-field ERG findings in the left eye were comparable to those in the right eye. On pattern ERG of the left eye, a decreased amplitude of the P50 was observed, while the N95 amplitude was unaffected. On multifocal ERG, the lowering of the N1P1 amplitude was found in the nasal quadrants (Figure 3).

The analysis of the clinical data, with a pivotal role of additional imaging, has led to the diagnosis of AZOOR. As there was no foveal involvement and normal visual acuity was preserved, there were no indications for pharmacological treatment. The patient remains in a semiannual follow-up at our department, lasting over five years since the diagnosis, during which a thorough assessment with color fundus photography, SS-OCT, FAF, and ICGA is performed. During the follow-up, no substantial progression of the disorder has been observed. Findings for the left eye on all diagnostic modalities have remained stable during the follow-up (Figure 1B,D,F,H,J). Although valproate treatment was terminated after three years, the patient remains free of seizures.

## 3. Discussion

To the best of our knowledge, this is the first patient with AZOOR and concomitant epilepsy reported in the literature. To date, AZOOR has been associated with numerous diseases including herpes simplex dermatitis, central nervous system abnormalities, viral infections, migraines, tick bites, and autoimmune diseases (in 28% of cases). An association with pregnancy was also reported [1]. Certain epidemiologic and clinical features have been linked with AZOOR. The majority of patients are young white women with myopia and photopsias that can precede or follow visual field loss corresponding to the affected retina. Bilateral presentation has been documented in 76% of cases. However, asymmetry is usually observed [1,8]. As central vision is often spared, the visual acuity remains relatively unaffected. Even after treatment, the complete recovery from visual field defects is uncommon. One-fourth of patients develop relative afferent pupillary defect (RAPD) several weeks after onset [1]. Initially, patients exhibit no fundoscopic changes. Therefore, the diagnosis is often delayed and is established after fundus alterations occur in more advance stages of the disease [1,9]. Another key finding in AZOOR is zonal visual field defect [1]. Among the multiple patterns of visual field loss, the most prevalent is an enlarged blind spot with central scotoma [1,10]. Some related abnormalities, such as inflammation in the vitreous, perivascular exudates, cystoid macular edema, changes in the retinal pigment epithelium (RPE), narrowing of the retinal vessels, and optic nerve edema have also been reported [1,9]. Although the clinical appearance of AZOOR varies depending on disease duration and distribution of lesions, there are some typical features including the demarcating line of progression between the involved and uninvolved retina, the trizonal pattern of sequential involvement of the outer retina, retinal pigment epithelium, and choroid, as well as frequent zonal progression [11,12,13,14]. In our patient, the development of symptoms, distribution, and morphology of abnormalities mimicked those in previously reported cases [9,10,15].

As the results of some imaging modalities can be ambiguous, the ophthalmologic community recommends ERG as a diagnostic tool of choice in patients with suspected AZOOR, as the tracing is always abnormal [1,8,9,16,17]. It is often used in the monitoring of disease progression, as the level of dysfunction of the outer retinal layers and the photoreceptors correlates with the degree of visual field loss [1,9]. The overall prognosis is generally favorable since AZOOR is a self-limiting disorder in most cases. Usually, the extent of visual field loss stabilizes after two to six months following the index event [1,9,15]. The follow-up of our patient supports these data, since no disease progression was observed over five years without targeted medical treatment.

No consensus on the treatment of AZOOR has been reached. However, systemic corticosteroids as well as immunosuppressive, antiviral, and antibacterial drugs are regarded as the main options. The role of topical medications is still discussed, since only vitritis has been reported to resolve once the treatment is started [9]. Some authors reported cases of spontaneous remission without treatment [18], whereas others argue that treatment including systemic corticosteroids reduces visual field defects and SS-OCT changes [19]. In line with our observations, Ibironke et al. [9] reported that despite permanent visual field loss the disease tends to be self-limiting, and, in most cases, it stabilizes before disabling central losses occur. However, if any related systemic illnesses are present, they need to be addressed and adequately managed [9].

Numerous studies linked visual disturbances and structural changes of the different retinal layers with antiepileptic drugs, but the effects of seizures themselves were not assessed in detail [20,21,22,23]. Drug-associated eye changes have to be included in the differential diagnosis. In our case, the patient was on valproate therapy, which causes alterations in color vision testing and visual evoked potentials, but does not account for changes in the visual field.

Recent investigations have established a direct link between responses in rat retina and epileptic episodes [4]. Research in rodents revealed substantial changes on a cellular level both immediately and at delayed periods following seizures. Despite relatively unaltered cytoarchitecture, prominent changes in glial cell activity and their protein expression reflecting retinal inflammation were observed. The glial activation may be hampered by antibody binding proteins, thereby proving it to be a part of systemic autoimmune response [4]. Based on the assumption that patients show similar retinal changes following epileptic seizures as rodents, Ahl et al. [4] claimed that the assessment of the retina could become a novel marker of seizure burden. Considering the convincing literature data and our case report, we hypothesize that epilepsy, due to its impact on the retinal structure, may increase the risk of other retinal disorders. However, this hypothesis requires further studies.

## 4. Conclusions

In summary, our case demonstrates the necessity for a thorough and multidisciplinary approach to retinal diseases such as AZOOR, especially in patients with concomitant diseases, such as epilepsy. In the presence of nearsightedness, normal fundus appearance, and the presumed impact of antiepileptic treatment, the presence of AZOOR may be easily overlooked.

## Figures and Tables

**Figure 1 medicina-57-01276-f001:**
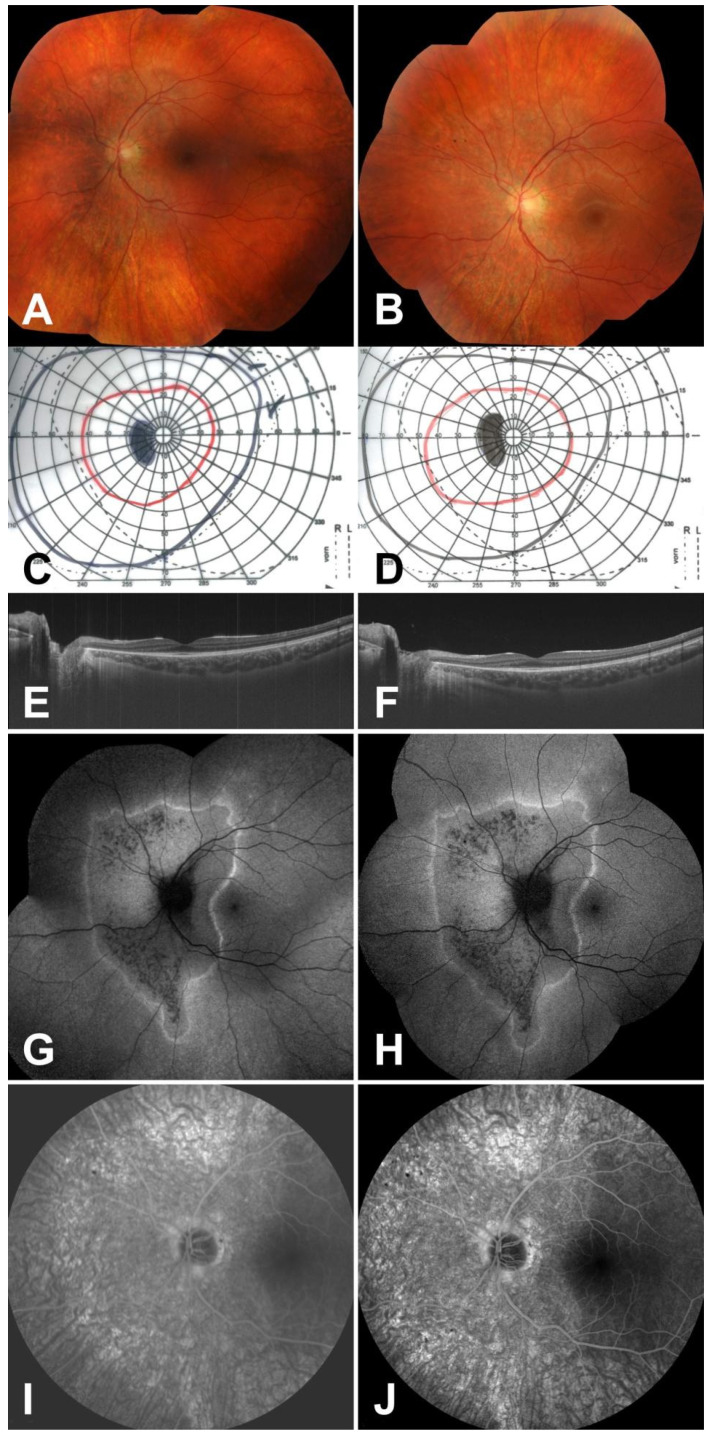
Changes in the affected eye on imaging tests on admission (**A**,**C**,**E**,**G**,**I**) and at five-year follow-up (**B**,**D**,**F**,**H**,**J**). Color fundus photography shows only an area of a thinner retina with some hyperpigmentation (**A**), as well as more pronounced hyperpigmentation and atrophic lesions after five years (**B**). Kinetic perimetry shows a slight enlargement of the blind spot (**C**,**D**). Swept-source optical coherence tomography reveals peripapillary disruption of the ellipsoid zone (**E**,**F**). Fundus autofluorescence shows an abnormal area encircled by a rim of high autofluorescence. The size of the spared perifoveal area remained stable on follow-up examinations (**G**,**H**). Fluorescein angiography (late phase) shows a peripapillary window defect of a similar extent comparing early and late phase (**I**,**J**).

**Figure 2 medicina-57-01276-f002:**
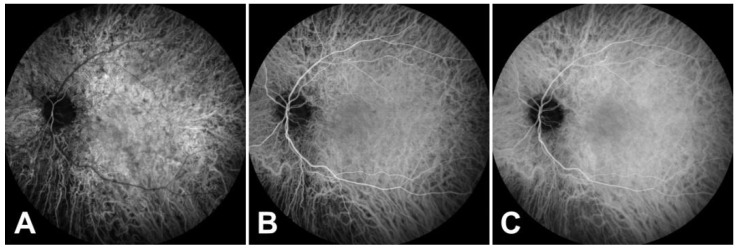
Indocyanine green angiography showing atrophy of the choriocapillaris in the affected area: early (**A**), middle (**B**), and late phases (**C**).

**Figure 3 medicina-57-01276-f003:**
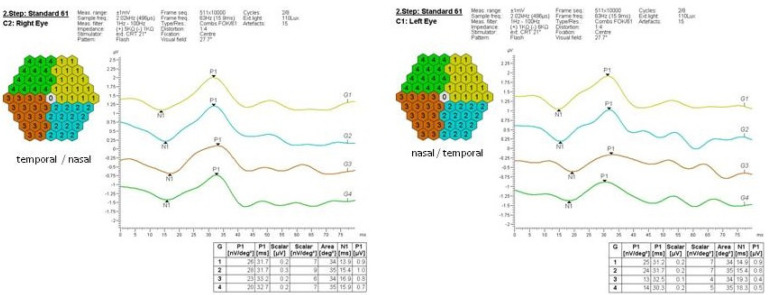
Multifocal electroretinography, amplitude N1P1, quadrant analysis. The deterioration of outer retinal layer function in the left eye is seen as the lower amplitude in the nasal quadrants.

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
