# Peer review of "Acute Zonal Occult Outer Retinopathy in a Patient Suffering from Epilepsy: Five-Year Follow-Up"

_medicina, 2021, doi:10.3390/medicina57111276_

Round 1

Reviewer 1 Report

Karska-Basta et al present a case report of a patient experiencing AZOOR coexisting with epilepsy. 

This is a well-written and well-presented case report.  The clinical functional and imaging studies are thorough.  The authors posit the potential for a link between epilepsy and retinal disorders.  My only criticism is that the authors put a lot of emphasis on a single study in rats and this case report. 

Eg. Line 46-48:  statement  that “there is sound evidence suggesting a possible causative link between epilepsy and retinal disorders” is overstated.  They cite a single paper that reported increased microglia in the retina of rats that had electrically induced seizures

Similarly the last sentences 167-169 place to much emphasis of the single study in rats and their case report to build a link.

Overall though the idea warrants further epidemiological investigation.

Author Response

Reviewer 1.

Karska-Basta et al present a case report of a patient experiencing AZOOR coexisting with epilepsy.

This is a well-written and well-presented case report.  The clinical functional and imaging studies are thorough.  The authors posit the potential for a link between epilepsy and retinal disorders.  My only criticism is that the authors put a lot of emphasis on a single study in rats and this case report.

Thank you for this valuable comment. We agree that a single study on rats is not enough to provide enough evidence for a link between epilepsy and retinal disorders. The problem with rare disorders is that studies on larger populations are not feasible. We revised the relevant fragments in response to your critical remarks.

Eg. Line 46-48:  statement  that “there is sound evidence suggesting a possible causative link between epilepsy and retinal disorders” is overstated.  They cite a single paper that reported increased microglia in the retina of rats that had electrically induced seizures

Thank you. We revised the statement.

Similarly the last sentences 167-169 place to much emphasis of the single study in rats and their case report to build a link.

The statement was revised.

Overall though the idea warrants further epidemiological investigation.

The statement was revised.

Reviewer 2 Report

I read the paper entitled “Acute Zonal Occult Outer Retinopathy in a Patient Suffering from Epilepsy” very carefully and concluded that the paper is acceptable with some revision for publication in your journal. The topic of the article is interesting, because the cases of AZOOR are rare. AZOOR was often misdiagnosed in the past as optic neuritis or pseudobilateral hemianopsia mimicking chiasma compression.

Some corrections must be made by the authors.

Some references regarding AZOOR must be added from the last years (Karagianis 2014, Matsui 2024, Aleman 2017, Boudreault 2017, Qian 2017, Duncker 2018, Introini 2018, Naik 2’18, Wang 2018, Rishi 2019, Lee 2020, Peng 2020, Herbort 2021).

The refractive error of the patient must be added, because the majority of AZOOR patients are myopic.

The patient was on Valproat therapy. How long was the patient on therapy? This anti-epileptic medication can cause visual changes as side effects of therapy (Totan 2017, Guimaraes-Souza 2018, Mattheo 2019). That must be involved in the discussion part and the authors must distinguish Valproat therapy influence from AZOOR..

Author Response

I read the paper entitled “Acute Zonal Occult Outer Retinopathy in a Patient Suffering from Epilepsy” very carefully and concluded that the paper is acceptable with some revision for publication in your journal. The topic of the article is interesting, because the cases of AZOOR are rare. AZOOR was often misdiagnosed in the past as optic neuritis or pseudobilateral hemianopsia mimicking chiasma compression.

Some corrections must be made by the authors.

Some references regarding AZOOR must be added from the last years (Karagianis 2014, Matsui 2024, Aleman 2017, Boudreault 2017, Qian 2017, Duncker 2018, Introini 2018, Naik 2018, Wang 2018, Rishi 2019, Lee 2020, Peng 2020, Herbort 2021)

Thank you very much for you comment. We agree that some of these papers are relevant to our paper, and we decided to add the following to our reference list: Karagianis 2014, Boudreault 2017, Duncker 2018, and Herbort 2021.

The refractive error of the patient must be added, because the majority of AZOOR patients are myopic.

We are grateful for this remark. This association is also true in our case: RE -4.25/-0.25 ax 150; LE -2.0/-0.25 ax 173. This was added in the paper.

The patient was on Valproat therapy. How long was the patient on therapy? This anti-epileptic medication can cause visual changes as side effects of therapy (Totan 2017, Guimaraes-Souza 2018, Mattheo 2019). That must be involved in the discussion part and the authors must distinguish Valproat therapy influence from AZOOR..

Thank you for this highly relevant question. Indeed, this distinction can be of high clinical value for readers. In our paper, we mentioned drug reactions as one of the differentials, but we agree that we need to elaborate on this. Our patient remained on Valproate treatment for 4 years. Valproate therapy can cause alterations in color vision testing and visual evoked potentials, but does not account for changes in the visual field. Therefore, in our case, drug-related changes were excluded. We also added appropriate references (Hilton 2004, Matthew 2019, and Ozkul 2002).

Reviewer 3 Report

I have no objections to the case report, in fact, I read it with great interest and learned something new!

Author Response

I have no objections to the case report, in fact, I read it with great interest and learned something new!

Thank you for the positive opinion.

English language corrections have been made